# Hierarchical morphogenesis of swallowtail butterfly wing scale nanostructures

Kwi Shan Seah[1,2], Vinodkumar Saranathan[1,2,3,4,5]*

[1]Division of Science, Yale-NUS College, Singapore, Singapore; [2]Department of Biological Science, National University of Singapore, Singapore, Singapore; [3]NUS Nanoscience and Nanotechnology Initiative (NUSNNI-NanoCore), National University of Singapore, Singapore, Singapore; [4]Lee Kong Chian Natural History Museum, National University of Singapore, Singapore, Singapore; [5]Present Address: Division of Sciences, School of Interwoven Arts and Sciences, Krea University, Central Expressway, Sri City, India

**Abstract** The study of color patterns in the animal integument is a fundamental question in biology, with many lepidopteran species being exemplary models in this endeavor due to their relative simplicity and elegance. While significant advances have been made in unraveling the cellular and molecular basis of lepidopteran pigmentary coloration, the morphogenesis of wing scale nanostructures involved in structural color production is not well understood. Contemporary research on this topic largely focuses on a few nymphalid model taxa (e.g., *Bicyclus*, *Heliconius*), despite an overwhelming diversity in the hierarchical nanostructural organization of lepidopteran wing scales. Here, we present a time-resolved, comparative developmental study of hierarchical scale nanostructures in *Parides eurimedes* and five other papilionid species. Our results uphold the putative conserved role of F-actin bundles in acting as spacers between developing ridges, as previously documented in several nymphalid species. Interestingly, while ridges are developing in *P. eurimedes*, plasma membrane manifests irregular mesh-like crossribs characteristic of Papilionidae, which delineate the accretion of cuticle into rows of planar disks in between ridges. Once the ridges have grown, disintegrating F-actin bundles appear to reorganize into a network that supports the invagination of plasma membrane underlying the disks, subsequently forming an extruded honeycomb lattice. Our results uncover a previously undocumented role for F-actin in the morphogenesis of complex wing scale nanostructures, likely specific to Papilionidae.

## eLife assessment

This **important** study reports how swallowtail butterflies pattern structures composed of chitin at the nanometer scale to produce structural colors. The work uses state-of-the-art microscopy techniques to **convincingly** show that F-actin is utilized in these butterflies in a novel way to produce structure, paving the way for further studies on growth regulation leading to precise ultrastructures and structural colors.

## Introduction

The patterning and coloration of butterfly wings have been a paradigmatic research focus due to their fundamental role in signaling and crypsis (*Beldade and Brakefield, 2002*; *Sweeney et al., 2003*; *Tsai et al., 2020*). Significant advances have been made recently in identifying the cellular and molecular basis of lepidopteran pigmentary coloration (*Matsuoka and Monteiro, 2018*; *Nadeau et al., 2016*; *Nishikawa et al., 2015*; *Reed et al., 2011*; *Van Belleghem et al., 2023*; *Zhang et al., 2017a*). A

*For correspondence:
vinodkumar.saranathan@aya.yale.edu

Competing interest: The authors declare that no competing interests exist.

small number of regulatory genes have been found to exert significant influence on the synthesis and spatial expression of pigments, as well as regulating cuticle deposition, thereby affecting the overall scale morphology (e.g., *Banerjee and Monteiro, 2022*; *Brien et al., 2022*; *Ficarrotta et al., 2022*; *Matsuoka and Monteiro, 2018*; *Prakash et al., 2022*; *Van Belleghem et al., 2023*; *Wee et al., 2022*). For instance, suppression of the *optix* gene has been found to tune lower lamina thickness, inducing iridescent structural coloration and affecting cuticular coverage of the upper lamina (*Wasik et al., 2014*; *Zhang et al., 2017b*).

Building on the classic studies on cellular organization of lepidopteran scales (*Ghiradella, 1974*; *Ghiradella, 1985*; *Ghiradella, 1989*; *Ghiradella and Radigan, 1976*; *Overton, 1966*), a few recent studies utilized advances in microscopy techniques to interrogate the formation of longitudinal ridges (*Day et al., 2019*; *Dinwiddie et al., 2014*; *McDougal et al., 2021*). These insights are, however, limited to ridge formation on the scale surface. Moreover, contemporary research on scale cell development focuses on just a few model taxa (*Bicyclus*, *Precis*, *Heliconius*, *Vanessa*) all from the family of Nymphalidae (*Beldade and Brakefield, 2002*; *Brien et al., 2022*; *Day et al., 2019*; *Dinwiddie et al., 2014*; *Matsuoka and Monteiro, 2018*; *Nadeau et al., 2016*; *Pomerantz et al., 2020*; *Prakash et al., 2022*; *Reed et al., 2011*; *Sweeney et al., 2003*; *Van Belleghem et al., 2023*; *Wasik et al., 2014*; *Zhang et al., 2017b*), despite an overwhelming diversity in the hierarchical organization of scale nanostructures across Lepidoptera (*Ghiradella, 1984*; *Ghiradella, 1985*; *Ghiradella, 1989*; *Prum et al., 2006*). A rare exception is the recent work on pierid butterfly *Colias* identifying *bric à brac* as a genetic switch that suppresses UV-reflecting Christmas tree-like ridge gratings in males (*Ficarrotta et al., 2022*), even though the precise development of the ridge grating is still unresolved (*Ghiradella, 1974*). Deciphering the cellular and developmental basis of scale organization in beyond just a few model lepidopterans is also highly relevant to current challenges in the synthesis of complex submicron and micron-scale hierarchical nanostructures, and could inspire novel biomimetic routes to fabricate multifunctional nanomaterials (*Kolle et al., 2010*; *McDougal et al., 2019*; *Pokroy et al., 2009*; *Potyrailo et al., 2007*; *Siddique et al., 2017*; *Wilts et al., 2019*).

The *bauplan* of lepidopteran wing scales consists an ornamented upper lamina over a relatively unstructured basal lamina, supported by arches with pillar-like struts called trabeculae (*Ghiradella, 1984*; *Figure 1—figure supplement 1*). The upper lamina is comprised of longitudinal ridges with transverse crossribs framing a set of rectilinear windows. These windows typically open into the interior lumen of the scale cell, but can also be covered by a thin layer of cuticular lamina (*Ghiradella, 1984*). The sides of the ridges feature microribs – fine flute-like stripes visible at higher magnifications under an electron microscope (*Figure 1—figure supplement 1*).

Among all Lepidoptera families, swallowtail butterflies (Papilionidae) not only encompasses the known diversity of scale nanostructure, but also exhibit some of the most complex hierarchical

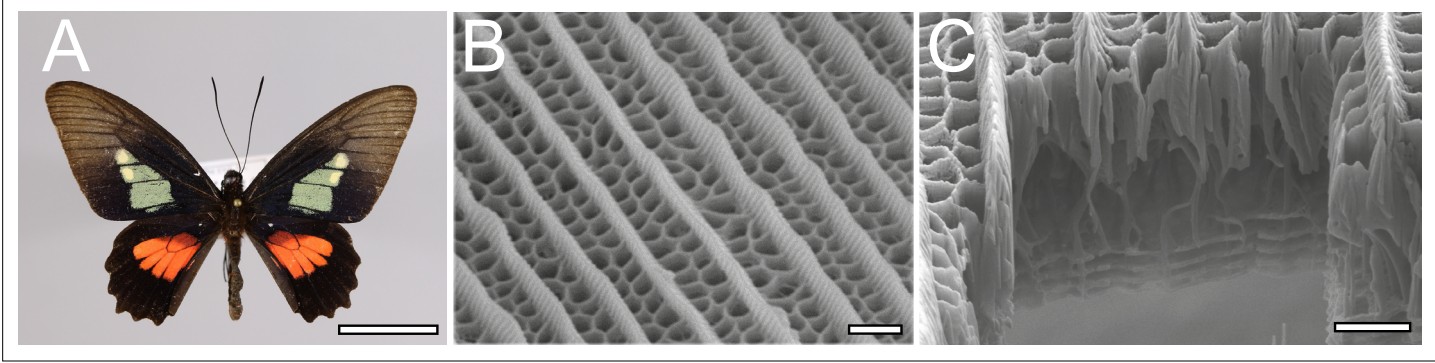

**Figure 1.** Hierarchical nanostructure of wing scales in *Parides eurimedes* (Papilionidae). (**A**) Adult male with structurally colored green patches on their dorsal forewings. Credit: iDigBio YPM Ent 433579. (**B**) Scanning electron microscopy (SEM) top view and (**C**) Focused ion beam (FIB)-SEM cross-sectional image of an adult green scale showing ridges and honeycomb lattice ending in trabeculae on top of a perforated multilayer lattice. Scale bars: (**A**) 1 cm, (**B, C**) 1 μm.

The online version of this article includes the following figure supplement(s) for figure 1:

**Figure supplement 1.** Diversity of adult wing scale nanostructure in Papilionidae shown in comparison to (R) a dorsal forewing (DFW) black scale of *Hypolimnas bolina* (Nymphalidae) with regular rectilinear crossribs.

morphologies found in nature (*Ghiradella, 1984*; *Ghiradella, 1985*; *Ghiradella, 1989*; *Prum et al., 2006*; *Saranathan et al., 2010*; *Figure 1—figure supplement 1*). Despite having a few model species (e.g., *Papilio xuthus*, *Papilio polytes*, and *Papilio machaon*), the focus of the current research on papilionids is limited to the molecular basis of mimicry of pigment patterns (e.g., *Nishikawa et al., 2015*; *Yoda et al., 2021*). The wing scales of papilionid species (e.g., *Parides sesostris*, *Papilio nireus*) exhibit irregular veins or mesh-like crossribs, often with an underlying honeycomb-like lattice of cuticular walls enclosing columnar pores (*Ghiradella, 1984*; *Ghiradella, 1985*; *Ghiradella and Radigan, 1976*; *Prum et al., 2006*; *Saranathan et al., 2010*; *Wilts et al., 2014*; *Figure 1* and *Figure 1—figure supplement 1*). In the following, we use 'honeycomb lattices' to refer to irregular crossribs with lattice walls extending into the interior lumen (e.g., green scales in *P. sesostris*, *Parides arcas*, and *Parides eurimedes*), and 'irregular crossribs' for crossribs that lack such extruded cuticular walls. In any case, these irregular crossribs are an autapomorphy of papilionid butterflies (*Ghiradella, 1984*; *Ghiradella, 1985*; *Ghiradella and Radigan, 1976*) that are thought to increase the surface area over which pigments such as papiliochromes can be deposited (*Wilts et al., 2012*), and likely play a role in thermoregulation as they are efficient absorbers of solar radiation (*Siddique et al., 2017*).

Here, we use scanning electron microscopy (SEM), conventional confocal and super-resolution structured illumination microscopy (3D-SIM) to study the time-resolved development of hierarchical scale nanostructures in wing scales of six papilionid species, focusing mainly on *P. eurimedes*. We discovered that following ridge formation, rows of chitinous disks form in between the ridges, followed by disintegration of F-actin that reorganizes into a mesh-like network. This network likely acts as a template for subsequent formation of honeycomb lattices by driving plasma membrane invagination into the scale lumen. Our study uncovers a novel process of wing scale nanostructure formation in papilionid species.

## Results

### Wing scale ridge formation is conserved in Papilionidae

We comparatively studied the wing scale growth in *P. polytes* and *P. eurimedes*, which have irregular crossribs and honeycomb lattices, respectively. Early stages of scale cell growth are as previously documented in several nymphalid species (*Day et al., 2019*; *Dinwiddie et al., 2014*; *Ghiradella, 1974*; *Pomerantz et al., 2020*). Wheat germ agglutinin (WGA) stains the plasma membrane during early stages of scale development in *P. eurimedes*. Scale cells from relatively young pupae at 38% development resemble elongated buds containing densely packed polymerizing F-actin filaments (*Figure 2—figure supplement 1A–A'' and B–B''*). At 43% development, F-actin filaments form thicker bundles that extend down the full length of scale cells (*Figure 2—figure supplement 1C'*), laying down a scaffold that determines the eventual position of ridges (*Dinwiddie et al., 2014*). WGA stains pleating membranes (longitudinal striations) in between adjacent rows of F-actin bundles (*Figure 2—figure supplement 1C–C'' and D–D''*). Around 48–50% development, the developing ridges can be more clearly discerned in between F-actin bundles (*Figure 2—figure supplement 1E–E''* and *Figure 4—figure supplement 6D–D'' and E-E''*). At this stage, there also appear to be irregular gaps in WGA staining in between the ridges of *P. eurimedes* scales (*Figure 2—figure supplement 1E*). These vein-like features resemble the irregular crossribs seen in adult wing scales (*Figure 1B and C* and *Figure 4—figure supplement 6B and C*).

### Crossrib and cuticular disk formation in *P. eurimedes*

We use lectin and membrane stains to follow plasma membrane topology and cuticle deposition during scale cell maturation (*Figure 2* and *Figure 2—figure supplements 2–5*). At 48% development, plasma membrane appears mottled while WGA predominantly stains longitudinal striations that will become ridges (*Figure 2A–C* and *Figure 2—figure supplement 2B–B''*). In cross-sections, both stains appear to be in close contact at the periphery of the cell, with plasma membrane underlying the cuticular layer (*Figure 2C* and *Figure 2—figure supplement 2C–C''*). This observation is expected as the cuticle of extracellular origin is being deposited onto the scale cell membrane. At 52% pupal development, plasma membrane shows distinct pleating corresponding to the developing ridges, with irregular crossribs in between (*Figure 2A'–C'* and *Figure 2—figure supplement 3A'–C'*). The bulk of cuticle remains above the plasma membrane at this stage (*Figure 2C'* and *Figure 2—figure*

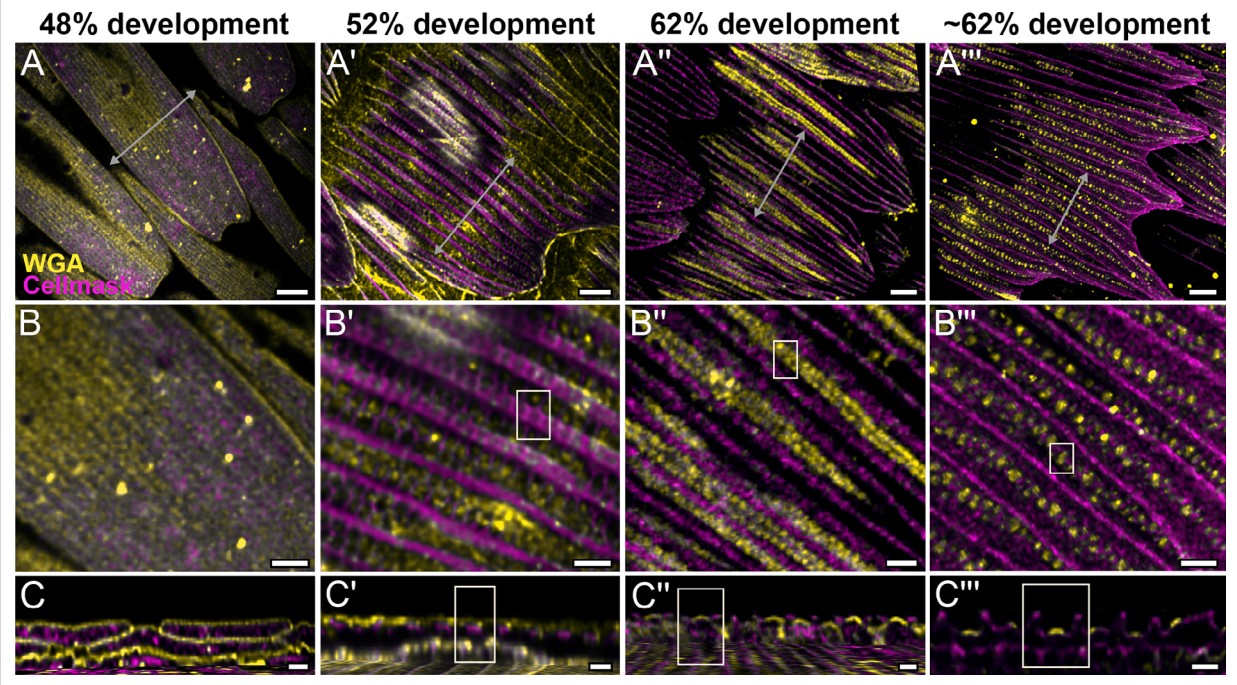

**Figure 2.** Morphogenetic time series of the development of cuticular disks in pupal *P. eurimedes* dorsal forewing scales, acquired with a 100× confocal microscope. AF-555 wheat germ agglutinin (WGA) (yellow) stains chitin and CellMask (magenta) stains plasma membrane. By 52% development, hollow vein-like crossribs appear on plasma membrane in between ridges that serve as a scaffold for cuticle accretion. As scales mature, more cuticle is deposited into these rows of disks bounded by plasma membrane. See also *Figure 2—figure supplements 3–5*. (**B–B'''**) Closeup views of (**A–A'''**). (**C–C'''**) *xz* cross-sections of the scale at locations marked with gray lines in (**A–A'''**) reveal the planar aspect of the cuticular disks. Yellow regions of interest (ROIs) in (**C'–C'''**) correspond to those in (**B'–B'''**). Scale bars (**A–A'''**) 5 µm, (**B–B''' and C–C'''**) 2 µm.

The online version of this article includes the following figure supplement(s) for figure 2:

**Figure supplement 1.** Early development of dorsal forewing green cover scales in pupal male *P. eurimedes* acquired with 60× confocal microscope.

**Figure supplement 2.** Yellow dorsal forewing cover scales of pupal female *P. eurimedes* at 48% development acquired with 100× confocal microscope.

**Figure supplement 3.** Yellow dorsal forewing cover scales of pupal female *P. eurimedes* at 52% development acquired with 100× confocal microscope.

**Figure supplement 4.** Yellow dorsal forewing cover scales of pupal female *P. eurimedes* at 62% development acquired with 100× confocal microscope.

**Figure supplement 5.** Black dorsal forewing scales of pupal male *P. eurimedes* at approximately 62% development acquired with 100× confocal microscope.

*supplement 3C''*). At around 62% pupal development, ridges appear to have grown to their near-final configuration while lectin staining reveals planar, disk-like cuticular features (hereafter cuticular disks) arranged in rows in between the ridges (*Figure 2A''–C'' and A'''–C'''* and *Figure 2—figure supplements 4 and 5*).

### Dynamics of F-actin reorganization in *P. eurimedes*

We also co-stain chitin and F-actin on the developing pupal scales to investigate the role of cytoskeleton in scale nanostructure formation. Interestingly, as cuticular disks of various sizes are forming in between ridges at 57% development, F-actin bundles are breaking down into short fibrils distributed around the disks (*Figure 3B–C*, *Figure 3—figure supplement 1*, and *Figure 3—video 1*). As the scales became more sclerotized and flattened out later in development (at 67%), disintegrating F-actin appears to reorganize and associate more clearly with cuticular disks, which now appear elongated and slightly tubular in cross-sectional view (*Figure 3A'–C'*, *Figure 3—figure supplement 2*, and *Figure 3—video 2*). By 76% development, the cuticular disks appeared hollow and are surrounded by distinct bulb-like structures formed by the disintegrated F-actin bundles (*Figure 3A''–C''*, *Figure 3—figure supplement 3*, and *Figure 3—video 3*).

To better understand the mechanism behind F-actin reorganization, we investigated whether the actin-related proteins, Arp2/3 complex, is a potential factor regulating the dendritic growth of actin

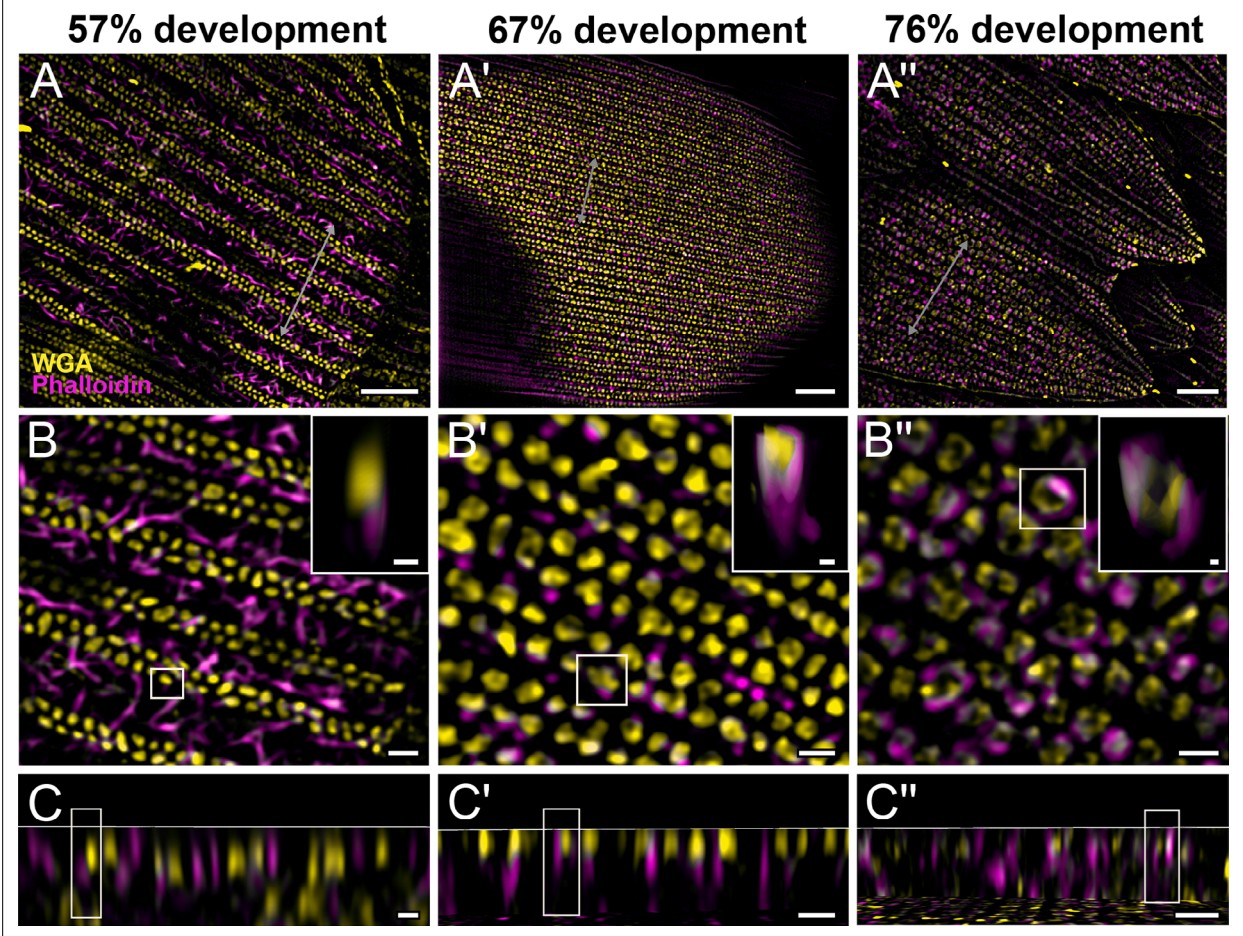

**Figure 3.** Morphogenetic time series of the development of columnar honeycomb lattice in pupal *P. eurimedes* dorsal forewing scale cells acquired with super-resolution lattice structured illumination microscopy (SIM). AF-555 wheat germ agglutinin (WGA) (yellow) stains chitin and AF-647 phalloidin (magenta) stains F-actin. AF-555 WGA shows a gradual evolution of the cuticular disks from filled-in planar to hollow tubular outgrowths. The disintegrating F-actin bundles show evidence of reorganization from linear to reticulated features with ring-like cross-sections. See also *Figure 3—figure supplements 1–3*. (**B–B''**) Closeup views of (**A–A''**). Insets correspond to the regions of interest (ROI) marked in yellow shown with a 3D aspect. (**C–C''**) *xz* cross-sections of the scale at locations marked with gray lines in (**A–A''**). Scale bars (**A–A''**) 5 µm, (**B–B''**, **C–C''**, and insets) 1 µm.

The online version of this article includes the following video and figure supplement(s) for figure 3:

**Figure supplement 1.** Green dorsal forewing cover scales of pupal male *P. eurimedes* at 57% development acquired with super-resolution lattice structured illumination microscopy (SIM).

**Figure supplement 2.** Green dorsal forewing cover scales of pupal male *P. eurimedes* at 67% development acquired with super-resolution lattice structured illumination microscopy (SIM).

**Figure supplement 3.** Black dorsal forewing scales of pupal male *P. eurimedes* at 76% development acquired with super-resolution lattice structured illumination microscopy (SIM).

**Figure 3—video 1.** Planar sections through the same dataset (57% development) as depicted in *Figure 3—figure supplement 1*.
https://elifesciences.org/articles/89082/figures#fig3video1

**Figure 3—video 2.** Planar sections through the same dataset (67% development) as depicted in *Figure 3—figure supplement 2*.
https://elifesciences.org/articles/89082/figures#fig3video2

**Figure 3—video 3.** Planar sections through the same dataset (76% development) as depicted in *Figure 3—figure supplement 3*.
https://elifesciences.org/articles/89082/figures#fig3video3

fibrils (*Hudson and Cooley, 2002*; *Pollard, 2007*; *Rouiller et al., 2008*). At around 52% development in *P. eurimedes*, Arp2/3 complex appear in a sparse punctate pattern while the F-actin bundles are still intact (*Figure 4A–C* and *Figure 4—figure supplement 1*). As F-actin bundles disintegrate and start to reorganize around 62% development, a relatively higher density of punctate Arp2/3 complex is seen

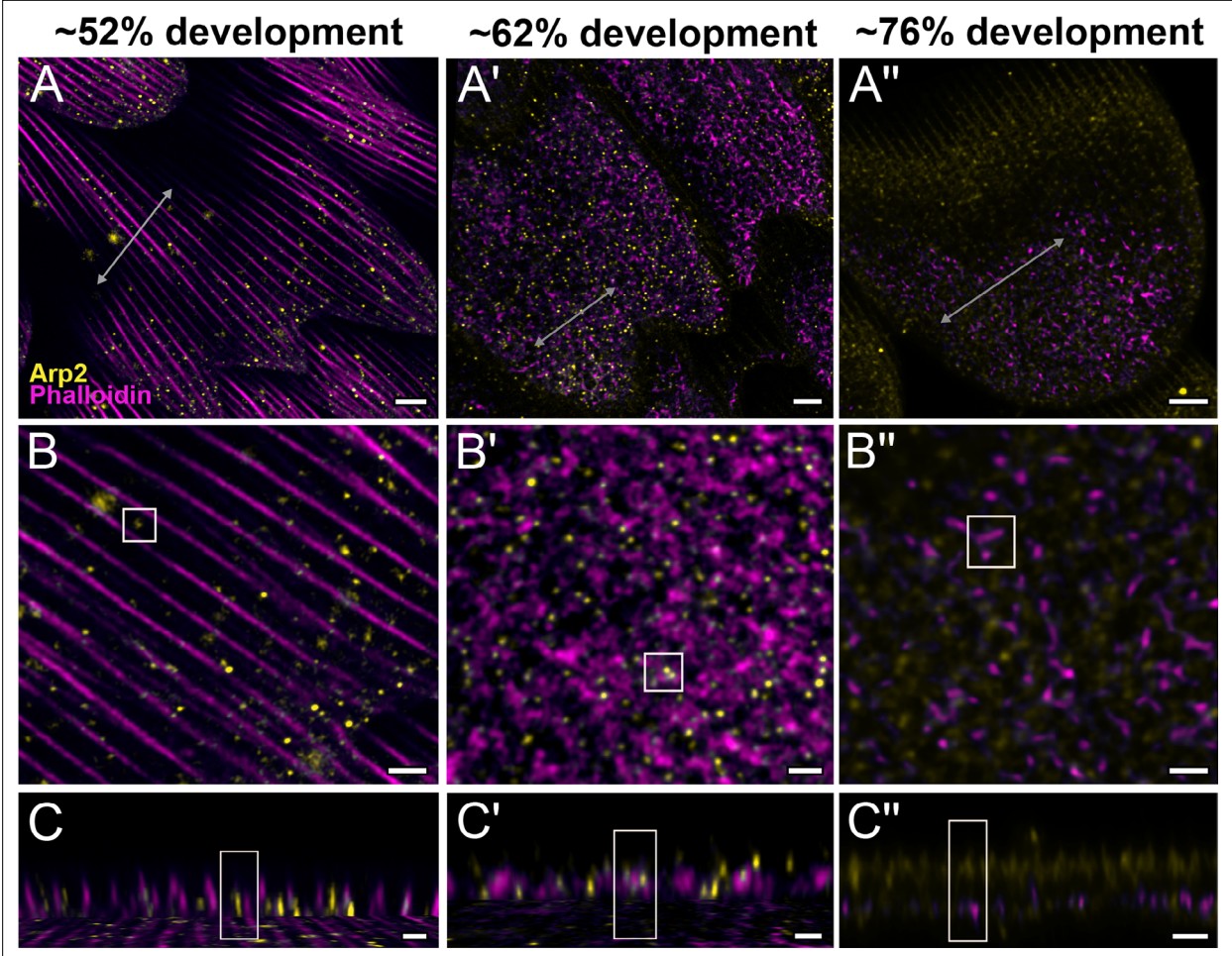

**Figure 4.** Arp2/3 complex is involved in F-actin reorganization in pupal *P. eurimedes* wing scales. Dorsal forewing scales stained with AF-594 anti-Arp2 (yellow) and AF-647 phalloidin (magenta) acquired with 100× confocal microscope. Initially (~52% development), Arp2/3 complex appear as sparse punctate dots while the F-actin bundles are still intact. As F-actin bundles disintegrate and reorganize, a relatively higher density of punctate Arp2/3 signal is seen in close association with the reticulate F-actin network. Our oldest timepoint in *P. eurimedes* (~76% development) has a large amount of cuticle autofluorescence overlapping with the AF594 signal, but the punctate pattern can be still discerned. See also *Figure 4—figure supplements 1–3*. (**B–B''**) Closeup views of (**A–A''**). (**C–C''**) *xz* cross-sections of the scale at locations marked with gray lines in (**A–A''**). Yellow regions of interest (ROIs) correspond to those in (**B–B''**). Scale bars (**A–A''**) 5 µm, (**B–B''** and **C–C''**) 2 µm.

The online version of this article includes the following source data and figure supplement(s) for figure 4:

**Figure supplement 1.** Yellow dorsal forewing cover scales of pupal female *P. eurimedes* at approximately 52% development acquired with 100× confocal microscope.

**Figure supplement 2.** Yellow dorsal forewing cover scales of pupal female *P. eurimedes* at approximately 62% development acquired with 100× confocal microscope.

**Figure supplement 3.** Green dorsal forewing cover scales of pupal male *P. eurimedes* at approximately 76% development acquired with 100× confocal microscope.

**Figure supplement 4.** Negative controls for Arp2 antibody staining.

**Figure supplement 5.** Pharmacological disruption of Arp2/3 with CK-666 in *Parides* pupae.

**Figure supplement 5—source data 1.** FIJI/ImageJ (version 2.1.0/1.53c) measurements of scanning electron microscopy (SEM) images of developing *Parides eurimedes* and *P. iphidamas* scales respectively used to generate the violin plots in *Figure 4—figure supplement 5A and D*, along with a summary of the pairwise *t*-tests.

**Figure supplement 6.** Development of dorsal forewing black cover scales in pupal *P. polytes* acquired with 100× confocal microscope.

**Figure supplement 7.** CK-666 inhibition does not affect F-actin structure in *P. polytes*.

**Figure supplement 8.** Scanning electron microscopy (SEM) micrographs of *P. polytes* pupal scales injected with CK-666 at 50 (**A**), 60 (**B**), and 70% (**C**) development.

in close association with the F-actin network (*Figure 4A'–C'* and *Figure 4—figure supplement 2*). At a later stage (~76% development), no distinctive punctate patterns can be discerned (*Figure 4A''–C''* and *Figure 4—figure supplement 3*), which is expected given that the reorganization of F-actin is nearly complete by this stage (see *Figure 3*). However, the scales are more mature at this point with prominent ridges, and autofluorescence signal from the cuticle has shifted into the green wavelengths (*Figure 4—figure supplement 4*), complicating the interpretation of the Arp2/3 immunofluorescence signal.

## Pharmacological disruption of Arp2/3 using CK-666

In order to mechanistically test the putative role of Arp2/3 in reorganizing F-actin during honeycomb lattice formation, we carried out pharmacological disruption experiments in pupae of both *Parides* and *Papilio* species using CK-666, a known inhibitor of the Arp2/3 complex (*Henson et al., 2015*). Unfortunately, we were only able to obtain a small number of *Parides* pupae (N = 15), which were already somewhat mature when they arrived, and further, a third of the pupae were dead upon arrival (*Supplementary file 1*). Honeycomb lattices can be seen in scales from emerged males of both *P. eurimedes* and *P. iphidamas* pupae treated with CK-666 (*Figure 4—figure supplement 5*). However, the height of the honeycombs in scales from pupae treated on 52% development was on average lower than the control (*Figure 4—figure supplement 5A and D*). Furthermore, in *P. eurimedes*, the height of the honeycombs in scales from pupae injected at 57% development was higher on average than those treated at 52% development, but indistinguishable from any controls (*Figure 4—figure supplement 5A*). These results are reminiscent of a previous study utilizing cytochalasin-D to inhibit linear actin polymerization (*Dinwiddie et al., 2014*) that found that once the pupae were past a certain developmental stage, ridge formation is no longer disrupted by the inhibitor. However, despite this intriguing trend in our results being consistent with a hypothesized role for Arp2/3, we require data with robust sample sizes and younger time points (i.e., 28–48% development) for *Parides* spp. in future experiments to validate the role of Arp2/3.

We also injected CK-666 into the wing pouches of the more readily available *P. polytes* pupae from 40 to 80% development to cover the possible timeframe during which Arp2/3 complex could be involved in F-actin reorganization. However, neither cuticular disks nor the dendritic reorganization of linear F-actin bundles are observed in *P. polytes*, whose adult scales have irregular crossribs without the deep lattice walls, that is, no honeycombs (*Figure 4—figure supplement 6B and C*). Shortly after ridge formation at 60% development (*Figure 4—figure supplement 6D–D''*), irregular crossribs are seen in between intact F-actin bundles (*Figure 4—figure supplement 6F–F''*). At 70% development, strong chitin autofluorescence signal is seen at the ridges (*Figure 4—figure supplement 6H–I*). Since F-actin reorganization is absent in this species, Arp2/3 inhibition did not disrupt ridge and crossrib formation in *P. polytes* as expected (*Figure 4—figure supplement 7*). Concurrent SEM imaging of pupal scales from the same samples shows that the formation of irregular crossribs is similarly not affected (*Figure 4—figure supplement 8*). Our results suggests the formation of cuticular disks and their subsequent extrusion by reorganizing F-actin is restricted to species with elaborate honeycomb lattice walls such as *P. eurimedes*.

## Honeycomb morphogenesis is conserved across Papilionidae

We also assayed the development of pupal wing scales of several other papilionid species: *P. arcas*, *P. nireus*, and *P. palinurus* (*Figure 5*), Although we were unable to obtain pupae with clearly marked pupation dates for these three species, we observed similar early stages of honeycomb development as seen in *P. eurimedes*. In adult male *P. arcas*, green scales have well-developed honeycomb lattice walls that sit above single gyroid crystals (*Figure 5A*). During *P. arcas* pupal scale development, both solid and hollow (annular) cuticular disks can be observed in between intact linear actin bundles (*Figure 5B–B' and C–C'*). The same is observed in *P. nireus*, where the upper lamina of adult blue scales consists of an elaborate honeycomb lattice with smaller sized pores (*Figure 5D–F*). In *P. palinurus* adults, green forewing cover scales feature large concave depressions in between the ridges (*Figure 5G*). Phalloidin staining of pupal *P. palinurus* scales reveals reorganization of longitudinal F-actin bundles into a shallow concave network (concentric rings in cross-section) (*Figure 5H and I* and *Figure 5—video 1*), which closely correspond to the cuticular dimples seen in adult green cover scales.

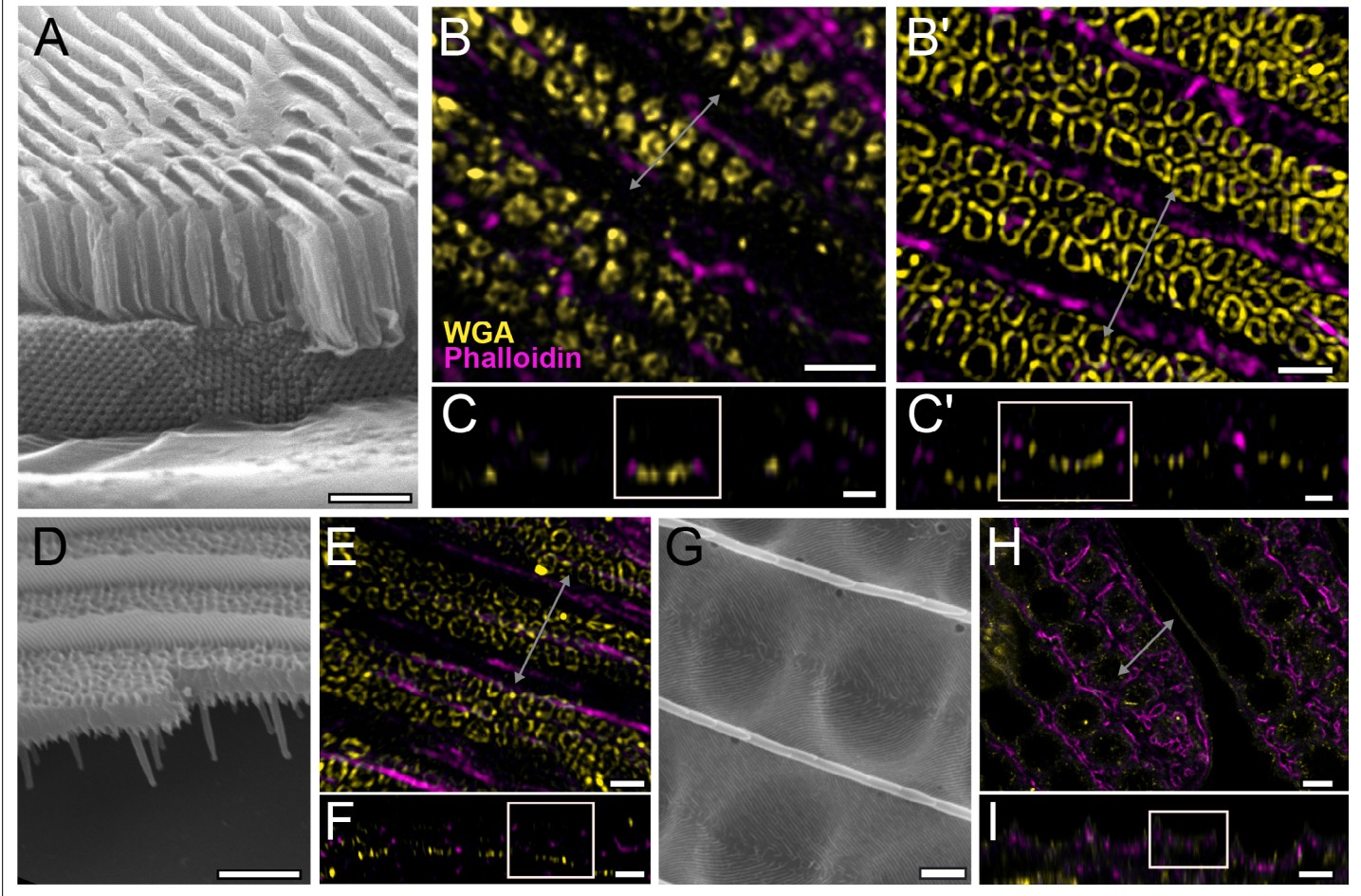

**Figure 5.** Conservation of honeycomb lattice development in Papilionidae. Scanning electron microscopy (SEM) cross-sectional view of (**A**) adult green scales in *P. arcas,* (**D**) blue scales in *P. nireus,* and (**G**) green scales in *P. palinurus*. All pupal dorsal forewing scales are stained either with AF-555 wheat germ agglutinin (WGA) or FITC-WGA (yellow) showing chitin, and AF-647 phalloidin or TRITC-phalloidin (magenta) showing F-actin. (**B, C**) Maximum projected 3D structured illumination microscopy (3D-SIM) micrograph of green cover scales in pupal male *P. arcas* featuring planar cuticular disks similar in shape and arrangement to *P. eurimedes*. (**B'–C'**) Maximum projected 3D-SIM micrograph of green cover scales of a different male *P. arcas* pupa showing irregular crossribs patterns. (**E, F**) Maximum projected 3D-SIM micrograph of blue cover scales of pupal *P. nireus*, similarly with crossribs and intact linear actin bundles. (**H, I**) Maximum projected 60× confocal micrographs of green cover scales of pupal *P. palinurus*. A concave network of F-actin underlies the cuticular dimples. At lower *z*, the actin rings (in cross-section) are smaller in size and show a foam-like appearance. (**C–C', F**, and **I**) *xz* cross-sections of the scales with region of interest (ROI) at locations marked with gray lines in (**B–B'**), (**E**), and (**H**), respectively. Scale bars (**A, D**, and **G**) 5 µm, (**B–B', C–C', E**, and **F**) 1 µm, (**H, I**) 5 µm.

The online version of this article includes the following video for figure 5:

**Figure 5—video 1.** Planar sections through the same dataset as depicted in *Figure 5H and I*, for *P. palinurus*.

https://elifesciences.org/articles/89082/figures#fig5video1

## Discussion

Swallowtail butterflies (Papilionidae) is the sister lineage to all other butterflies (*Espeland et al., 2018*) and the showcase family of lepidopterans as they exhibit some of the most diverse assortment of wing scale nanostructures (*Ghiradella, 1985*; *Prum et al., 2006*). Papilionid wing scales generally exhibit irregular crossribs, with some having underlying cuticular walls enclosing tubular pores, although the regularity and depth of the walls vary between species (*Figure 1—figure supplement 1*). These crossrib patterns are unlike the regular, rectilinear crossribs commonly seen in other lepidopteran families, for example, Nymphalidae (*Figure 1—figure supplement 1R*). In this study, we extended previous observations that parallel F-actin bundles configure the spacing and position of longitudinal ridges of wing scales (*Day et al., 2019*; *Dinwiddie et al., 2014*; *Ghiradella, 1974*; *Pomerantz et al.,*

*2020*) to Papilionidae, further supporting the hypothesis that this mechanism is broadly conserved across lepidopterans.

Our first novel finding is the appearance of irregular crossrib patterns in the plasma membrane between surface ridges, which seemed to serve as a template for chitin accretion into chitinous disks. This process is important as it marks the beginning of honeycomb lattice formation. Further, our results revealed that F-actin plays a predominant and previously unconsidered role in the morphogenesis of some papilionid butterfly wing scales. Once the longitudinal ridges have developed, F-actin bundles, which typically degenerate in nymphalid species (*Day et al., 2019*; *Dinwiddie et al., 2014*), subsequently reorganize into a reticulate network surrounding rows of cuticular disks on the scale surface. This actin network likely draws in plasma membrane underlying the cuticular disks into a porous honeycomb lattice. Similar nonlinear F-actin morphologies have been observed in other organisms. In diatoms, F-actin organizes into an interdigitating mesh-like porous network during development (*Tesson and Hildebrand, 2010*). This actin network defines frustule (cell walls) morphogenesis by providing a template for silica biomineralization at the meso- and microscales. In mammalian cells, transient ring-like F-actin structures are thought to drive autophagosome generation by serving as a scaffold for mitophagy initiation structures. 3D-SIM revealed F-actin partially associating with mitochondria in the form of curved sheet (*Hsieh and Yang, 2019*), akin to the F-actin structures seen in this study.

The schematic in *Figure 6* illustrates our proposed formation of honeycomb lattices in some papilionid wing scales. During early stages of scale formation, F-actin bundles prefigure the loci of future ridge formation in between adjacent actin bundles (*Dinwiddie et al., 2014*). Plasma membrane then forms a scaffold resembling the irregular crossrib pattern. The cuticle accretes into roughly circular disks delineated by the scaffold. Reorganization of F-actin around the chitinous disks draws the plasma membrane along with the overlying cuticle inward, molding it into the lengthened cuticular walls of the honeycomb lattice.

In our attempt to unravel the molecular players in actin reorganization, we used CK-666 to inhibit the role of Arp2/3 complex in actin branching (*Henson et al., 2015*). However, CK-666 did not quite disrupt the formation of honeycombs even when administered across multiple timepoints. A variety of reasons could potentially explain this. While 100 uM of CK-666 has been shown to work in other animal models (*Henson et al., 2015*), it might still not be a sufficient titer for papilionid butterfly pupal scales. Unfortunately, due to logistical difficulties in obtaining and rearing *Parides* spp., we were unable to perform a thorough optimization using CK-666. Furthermore, the *Parides* pupae were likely too mature at the time of CK-666 treatment, when the F-actin network has already been established (52% development). In a recent study (*Sakamoto et al., 2018*), CK-666 did not disrupt the formation of cortical actin meshwork in Sertoli cells of mouse seminiferous tubules as they were found to be regulated by formins rather than Arp2/3. This suggests that other molecular players, including formins, could be involved in wing scale honeycomb morphogenesis and need testing.

We note that the irregular crossrib patterns are in place while the ridges are still growing (*Figure 2A'–C'*). These crossribs appear to constrain and delineate the accumulation of cuticle into planar disks in between the ridges. A close examination of SEM images (*Figure 1B* and *Figure 1—figure supplement 1*) reveals that the endpoints of crossribs at the ridge interface are often connected to microribs present on the ridges. Given their relatively small size and pitch, we are unable to resolve the development of microribs here, which likely requires imaging techniques with even higher resolution such as photo-activated localization microscopy and stochastic optical reconstruction microscopy. However, both the periodic stripes (microribs) and quasi-periodic spots (crossribs) are reminiscent of Turing patterns (*Meinhardt, 1982*; *Turing, 1952*), that is, implying that an activator-inhibitor type mechanism could be involved in their formation. Alternatively, the crossrib pattern could be produced as a result of spontaneous processes like Ostwald ripening, phase separation, and/or biomechanical forces determined by the mechanical properties of the membranes, leading to stable, quasi-periodic, foam-like perforations on the plasma membrane in between ridges (*Dinwiddie et al., 2014*; *Elson et al., 2010*; *McDougal et al., 2021*; *Rosetti et al., 2017*; *Voorhees, 1985*; *Wilts et al., 2019*).

Overall, the presence of chitinous disks followed by F-actin reorganization appears to be specific to species with extended honeycomb lattice walls, for example, *P. eurimedes*, *P. arcas*, and *P. nireus* (*Figure 1—figure supplement 1A and E*). Such reorganized actin structures are not observed in species with shallow irregular crossribs, such as *P. polytes* (*Figure 4—figure supplement 6*). On the

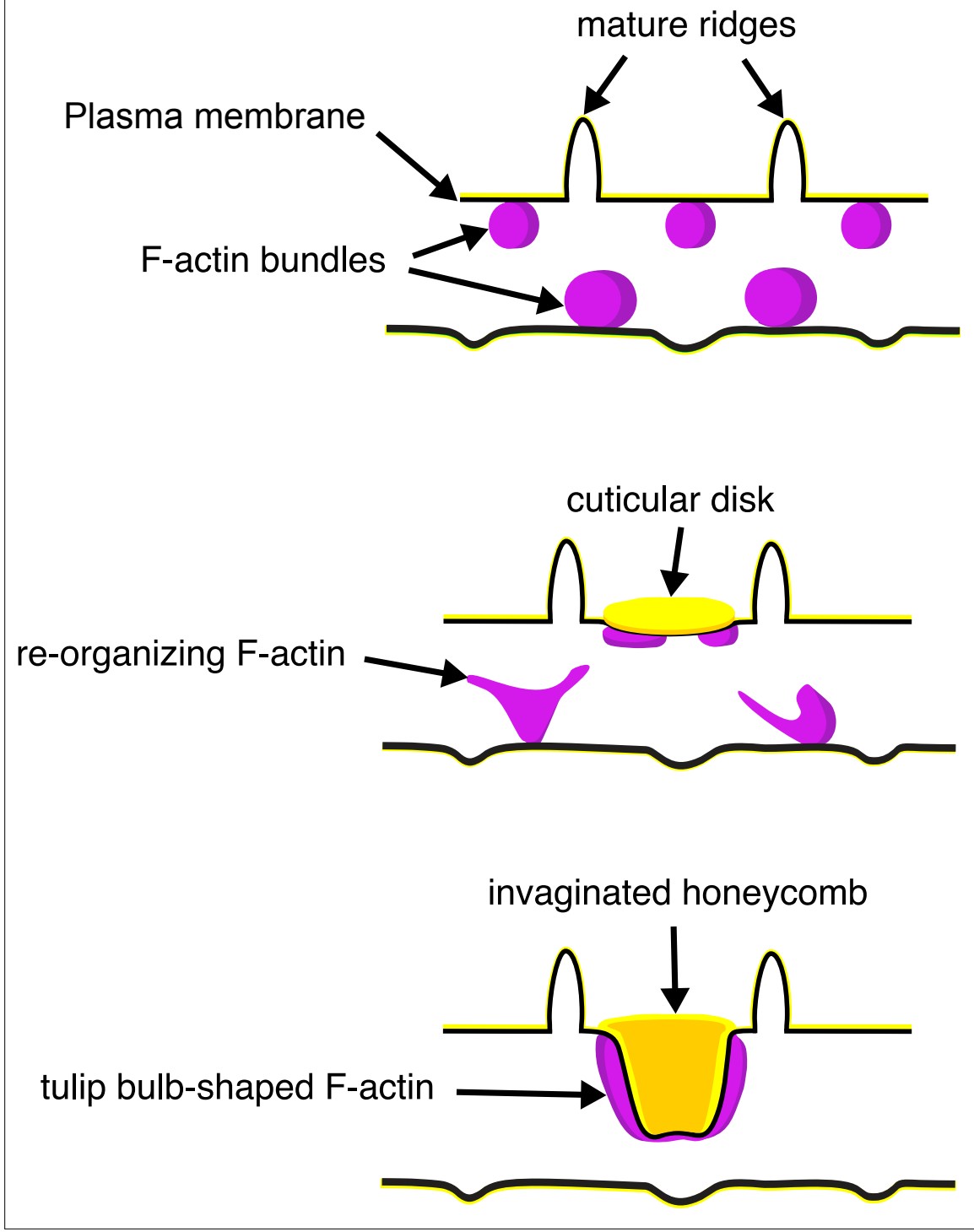

**Figure 6.** A schematic illustrating the proposed morphogenesis of honeycomb lattices in papilionid wing scales. During early stages, F-actin bundles prefigure the loci where ridges form in between adjacent actin bundles. Next the plasma membrane (black lines) forms a scaffold where cuticle (green lines and structures) accumulates into the spaces and forms irregular disk-like structures. Reorganization of F-actin around the disks subsequently extrudes the plasma membrane along with the deposited cuticle, forming the honeycomb lattice walls.

other hand, several *Papilio* species (e.g., *palinurus*, *blumei*, *karna*) possess multilayered cover scales (*Ghiradella, 1974*; *Prum et al., 2006*; *Trzeciak et al., 2012*) with widely spaced and reduced (in both height and number) ridges. These scales have no apparent crossribs, trabeculae, or honeycombs. Instead, they have a characteristic inter-ridge array of concave depressions and underlying perforated

lamellae in the scale interior (*Figure 1—figure supplement 1L and M*). Based on our observations, we speculate that the dimpled appearance of cover scales is templated by reorganization of F-actin bundles into a whorl-like concave network. This could represent an extreme modification of the proposed developmental program of the usual papilionid honeycomb lattice. Without crossribs, the pore sizes of honeycombs are likely constrained only by the pitch of the ridges. This is consistent with our observation that the outermost F-actin rings possess dimensions approaching the inter-ridge distance (*Figure 5C*). Without the trabeculae, the lumen multilayer fills the entire interior of the scale right up to the shallow dimples.

Interestingly, in typical lepidopteran scale cells, the trabeculae extend downward from the crossribs and form columns of arches in between the ridges (*Ghiradella, 1974*; *Ghiradella, 1985*; *Ghiradella, 1989*; *Ghiradella and Radigan, 1976*; *Overton, 1966*), suggesting that they are developmentally connected. Recently, *Matsuoka and Monteiro, 2018* found that *DDC* mutant of *Bicyclus anynana* (Nymphalidae) possessed irregularly spaced and thin crossribs with sheet-like vertical trabeculae instead of feet-like, arched trabeculae (see Figure 4A of *Matsuoka and Monteiro, 2018*). These mutant scale morphologies are somewhat reminiscent of the irregular crossribs and honeycomb lattice walls of Papilionidae. This suggests that spatio-temporal changes in expression patterns of single genes such as *DDC* could possibly alter crossrib morphology and drive honeycomb formation. However, any putative pleiotropic role of pigment-pathway genes in organizing papilionid scale morphology should be reconciled with that of actin reorganization. Future studies could look at knocking out *DDC* and other pigment-pathway genes during papilionid pupal development, in addition to inhibiting actin-binding proteins including Arp2/3 and formins, and tinkering with master regulatory genes like *optix* (*Banerjee and Monteiro, 2022*; *Wasik et al., 2014*; *Zhang et al., 2017b*).

The smooth endoplasmic reticulum (SER) has been implicated in templating luminal scale nano-structures during pupal development. Given that the papilionid honeycomb lattice extends into the lumen of the scale cells, any putative role of the SER in honeycomb morphogenesis should also be investigated. It would be of interest to follow the development of pupal wing scales using tissue clearing techniques or attempt a more finely resolved developmental time series to capture the full complexity of molecular and cytoskeletal dynamics. Comparatively understanding the morphogenesis of hierarchical nanostructures across biodiversity could inspire facile, biomimetic routes to synthesizing hierarchically structured materials for technological applications, given current challenges in the engineering of complex multifunctional mesophases (*Kolle et al., 2010*; *Pokroy et al., 2009*; *Potyrailo et al., 2007*; *Siddique et al., 2017*).

# Materials and methods

**Key resources table**

| Reagent type (species) or resource | Designation | Source or reference | Identifiers | Additional information |
|---|---|---|---|---|
| Strain, strain background (*Parides eurimedes*) | *P. eurimedes* | Stratford-upon-Avon Butterfly Farm (UK); Mariposario del Bosque Nuevo (Costa Rica) | | developing pupae |
| Strain, strain background (*Parides arcas*) | *P. arcas* | Stratford-upon-Avon Butterfly Farm (UK); Mariposario del Bosque Nuevo (Costa Rica) | | developing pupae |
| Strain, strain background (*Parides iphidamas*) | *P. iphidamas* | Mariposario del Bosque Nuevo (Costa Rica) | | developing pupae |
| Strain, strain background (*Papilio polytes*) | *P. polytes* | Marl Insect and Butterfly Culture (Philippines) | | developing pupae |

*Continued on next page*

*Continued*

| Reagent type (species) or resource | Designation | Source or reference | Identifiers | Additional information |
|---|---|---|---|---|
| Strain, strain background (*Papilio nireus*) | *P. nireus* | Stratford-upon-Avon Butterfly Farm (UK); Marl Insect and Butterfly Culture (Philippines) | | developing pupae |
| Strain, strain background (*Papilio palinurus*) | *P. palinurus* | Marl Insect and Butterfly Culture (Philippines) | | developing pupae |
| Antibody | Anti-Arp2 (rabbit) | Abcam | ab47654 | 1:500 |
| Antibody | Anti-rabbit Alexa Fluor 594 (goat) | Abcam | ab150088 | 1:300 |
| Antibody | Alexa Fluor 555-conjugated wheat germ agglutinin | Invitrogen | W32464 | 1:200 |
| Antibody | Alexa Fluor 647-conjugated phalloidin | Invitrogen | A22287 | 1:40 |
| Antibody | FITC-conjugated wheat germ agglutinin | EY-Labs | F-2101-5 | 1:100 |
| Antibody | TRITC-conjugated phalloidin | Sigma-Aldrich | P1951 | 1:100 |
| Chemical compound, drug | CellMask Deep Red Plasma Membrane Stain | Invitrogen | C10046 | 1:300 |
| Chemical compound, drug | CK-666 | Sigma-Aldrich | SML0006 | 100μM |
| Chemical compound, drug | ProLong Gold Antifade Mountant | Life Technologies | P36930 | |
| Software, algorithm | Imaris Viewer v9.5.1 | Bitplane AG | RRID:SCR_007370 | |
| Software, algorithm | Imaris v9.1 | Bitplane AG | RRID:SCR_007370 | |
| Software, algorithm | Shotcut v19.07.15 | Meltytech, LLC | | |
| Software, algorithm | Packages for R: *ggplot2*, base | R statistical environment (v4.1.2) | RRID:SCR_001905 | http://www.r-project.org |
| Software, algorithm | Huygens Professional v20.04 | Scientific Volume Imaging B.V. | | |
| Software, algorithm | FIJI/ImageJ (v2.1.0/1.53c) | 10.1038/nmeth.2019 | RRID:SCR_002285 | https://fiji.sc |

## Experimental design

To understand the process of scale nanostructure formation, we performed time-resolved imaging on developing pupal scales using specific biomarkers conjugated with fluorophores. Since cellular membranes and cytoskeletal elements have been identified as key components driving scale cell development, we used biomarkers targeting F-actin, cuticle (chitin), plasma membrane, and Arp2/3 complex. We used 3D-SIM and lattice SIM to try and resolve structures beyond the diffraction limit. We would have preferred to present all confocal and SIM data solely on a single patch, for instance, the green dorsal forewing cover scales of pupal *P. eurimedes*. However, for time points where scales were highly crumpled or folded, we addressed this gap with data from black areas adjacent to the green patch or from homologous yellow dorsal forewing patch on females. The overall consistency of the results validates our approach.

## Dissection of pupae and tissue preparation

Pupae were purchased in multiple batches from Marl Insect and Butterfly Culture (Philippines), Stratford-upon-Avon Butterfly Farm (UK), and Mariposario del Bosque Nuevo (Costa Rica) between June 2017 and April 2021. Precise pupation dates were only available for *P. polytes* and one batch of *P. eurimedes* pupae. For *Parides* batches without information on the pupation dates, we were able to estimate the rough pupation stage (indicated by ~, e.g., *Figure 2A'''–C''' and 4*) by morphological comparisons to this reference *P. eurimedes* batch. Sex of each pupa was determined by examining

markings on the ventral segments. Wings from each pupa were dissected in cold phosphate buffer saline (PBS) and immediately fixed in 4% PEM-PFA at room temperature for 15 min. Following washes in PBS, wings were placed in blocking buffer (0.5% NP-40) at 4°C overnight prior to staining. The average duration from pupation to emergence is 10 d for *P. polytes* and 21 d for *P. eurimedes*. We report the percentage of development in our time series for ease of comparison between species with different developmental rates.

### Developmental time series with wheat germ agglutinin, phalloidin, and anti-Arp2

Wheat germ agglutinin (WGA) was previously used to visualize butterfly scale cell growth and is thought to initially stain plasma membrane before switching to chitin at later developmental stages (*Dinwiddie et al., 2014*). Phalloidin is a standard method to visualize F-actin. For AF-555 WGA and AF-647 phalloidin double-staining, pupal wings were incubated in 1:200 dilution of Alexa Fluor 555-conjugated WGA (Invitrogen W32464) and 1:40 dilution of Alexa Fluor 647-conjugated phalloidin (Invitrogen A22287) for an hour at room temperature. For FITC WGA and TRITC phalloidin double staining, *P. palinurus* pupal wings were incubated in 1:100 dilution of FITC WGA (EY Labs F-2101-5) and 1:100 dilution of TRITC phalloidin (Sigma P1951) for an hour at room temperature. Arp2 is the ATP-binding component of the actin Arp2/3 complex, which functions as an actin nucleator in branched actin networks (*Goley and Welch, 2006*; *Smith et al., 2013*). After blocking, pupal wings were incubated with a 1:500 dilution of rabbit anti-Arp2 (Abcam ab47654; pblast search revealed UniProt #P61160, Human Arp2 has 82% sequence similarity to XP_013178655.1, *P. xuthus* Arp2) at 4°C overnight. After washing, the wings were incubated in buffer with a 1:300 dilution of Alexa Fluor 594 goat anti-rabbit secondary antibody (Abcam ab150088) for an hour at room temperature.

### Time series with CellMask plasma membrane stain

As the CellMask plasma membrane stain does not survive permeabilization, dissected pupal wings were immediately stained with a 1:300 dilution of CellMask Deep Red (Invitrogen C10046) for 10 min. After removing the staining solution, the wings were fixed in 4% PEM-PFA at room temperature for 15 min. Following washes in PBS, pupal wings were stained with a 1:200 dilution of Alexa Fluor 555-conjugated WGA (Invitrogen W32464) for an hour at room temperature. In order to prevent permeabilization, the buffers for CellMask stains did not contain any detergents (e.g., Triton).

### Arp2/3 inhibition with CK-666

CK-666 was purchased from Sigma-Aldrich (SML0006) and the stock solution was dissolved in DMSO. For pupal injections, CK-666 was diluted in 1× Grace's insect medium to final concentration of 100 µM. Then, 2–3 *P. polytes* pupae were injected with CK-666 at 40, 50, 60, 70, and 80% development. Around 8–10 µL of CK-666 was injected into wing pouches using a glass needle. Control pupae were injected with equal volume of DMSO diluted in Grace's insect medium. All injected pupae were dissected at 80% development and stained for immunofluorescence. Pupae injected at 80% development were allowed to develop for at least 2 hr before dissection.

### Negative controls

For experiments with non-fluorophore-conjugated primary antibodies (Arp2 staining), we labeled a separate wing from the same individual with the secondary antibody only. This is to determine whether the observed fluorescence signal is due to nonspecific binding of secondary antibodies. After blocking, pupal wings were incubated without primary antibodies (only buffer) at 4°C overnight. After washing, the wings were incubated with a 1:300 dilution of Alexa Fluor 594 goat anti-rabbit secondary antibody (Abcam ab150088) for an hour at room temperature. Negative controls were imaged using the same settings (laser power, gain, etc.) as the anti-*Arp2* antibody-stained test samples.

### Image acquisition and data processing

Following washes, the wings were mounted on glass slides in Prolong Gold antifade (Life Technologies P36930), covered with #1 thickness coverslips and sealed with nail polish. Confocal images were acquired using Olympus FV3000 (60×), Nikon A1R (100×), and Leica SP8 (100×). 3D-SIM was performed on a DeltaVision OMX and lattice SIM using Zeiss Elyra 7. Confocal data acquired with

Olympus FV3000 and Nikon A1R were deconvolved using default settings in Huygens Professional v20.04. Confocal data acquired with Leica SP8 were deconvolved using default settings in Leica's Lightning deconvolution software during acquisition. All images were examined and processed using Bitplane Imaris Viewer v9.5.1. Movies were generated using Bitplane Imaris v9.1 and edited using Shotcut v19.07.15 (Meltytech, LLC).

## Critical point drying (CPD)

Wings from each pupa were dissected in cold PBS and immediately fixed in 4% PEM-PFA at room temperature for 15 min. Following washes in PBS, wings were subjected to serial dehydration in 25, 50, 70, and 90% and absolute ethanol for 10 min each. The specimens were then transferred to the sample holders and dried using Leica EM CPD300 Automated Critical Point Dryer with the following settings: ($CO_2$ influx) medium speed, 120 s delay; (exchange process) speed 5, 14 cycles; medium heating speed; and medium gas out speed. It should be noted that direct air drying of soft samples, for example, pupal wing scales is not recommended, as water surface tension forces will result in significant distortion of specimen structure (*Bray et al., 1993*).

## Conventional and focused ion beam (FIB) SEM

Adult scales were individually picked with a needle and placed on carbon tape. Mounted scales were fractured using a razor-blade to obtain cross-sectional views. All samples were sputter-coated with platinum or gold to increase conductivity and reduce charging. Samples were imaged using JEOL JSM 6010LV Scanning Electron Microscope at 15–20k.

For FIB milling, samples were prepared by sputter-coating with platinum to increase conductivity. The sectioned scale shown in *Figure 1C* is milled using a gallium ion beam on a FEI Versa 3D with the following settings: beam voltage 8 kV, beam current 12 pA at a 52° tilt. Image acquisition was performed in the same equipment with the following settings: beam voltage -5 kV, beam current 13 pA.

## Statistical analyses of data from CK-666 inhibition experiments

One-sided ($H_a$: treatment < control) pairwise *t*-tests with default settings (pooled standard deviations, Holm method to correct for multiple comparisons) were performed in the *R* statistical environment (version 4.1.2) running on macOS Monterey (version 12.6.5).

## Acknowledgements

We are indebted to Antonia Monteiro for generously providing us with lab resources, including the use of insectary. We thank Cédric Finet for sharing FIB-SEM image of *P. eurimedes*. We are grateful to the Monteiro lab, C Finet, Cynthia He, Rong Li, Sasha Bershadsky, and Dan Morse for their thoughtful suggestions. We thank Sree Vaishnavi Sundararajan and Gianluca Grenci (MBI) for access and help with SEM, Tong Yan (CBIS), Mak Kah Jun and Peng Qiwen (MBI), Laura and Keshmarathy Sacadevan (Singhealth Advanced Bioimaging) for help with confocal microscopy, Graham Wright and Goh Wah Ing (A*STAR AMP) and Shi Xianke (Zeiss) for help with SIM data acquisition.

## Additional information

### Funding

| Funder | Grant reference number | Author |
|---|---|---|
| Society for the Study of Evolution | 2019 GREG Rosemary Grant Advanced Award | Kwi Shan Seah |
| Yale-NUS College | Startup funds: R-607-261-182-121 | Vinodkumar Saranathan |
| National Research Foundation Singapore | CRP20-2017-0004 | Vinodkumar Saranathan |

| Funder | Grant reference number | Author |
|---|---|---|

The funders had no role in study design, data collection and interpretation, or the decision to submit the work for publication.

## Author contributions

Kwi Shan Seah, Conceptualization, Data curation, Formal analysis, Investigation, Visualization, Methodology, Writing – original draft, Writing – review and editing; Vinodkumar Saranathan, Conceptualization, Supervision, Funding acquisition, Visualization, Methodology, Project administration, Writing – review and editing

## Author ORCIDs

Kwi Shan Seah (ID) https://orcid.org/0000-0001-8451-3732
Vinodkumar Saranathan (ID) https://orcid.org/0000-0003-4058-5093

Reviewer #1 (Public Review): https://doi.org/10.7554/eLife.89082.3.sa1
Reviewer #2 (Public Review): https://doi.org/10.7554/eLife.89082.3.sa2
Author Response https://doi.org/10.7554/eLife.89082.3.sa3

# Additional files

## Supplementary files

• Supplementary file 1. Details of CK-666 inhibition experiments performed in different batches of purchased pupae and their eventual fates.

• MDAR checklist

## Data availability

All data generated or analysed during this study are included in the manuscript and supporting files.

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
