## [Editor Report · eLife assessment]

This **important** study reports how swallowtail butterflies pattern structures composed of chitin at the nanometer scale to produce structural colors. The work uses state-of-the-art microscopy techniques to **convincingly** show that F-actin is utilized in these butterflies in a novel way to produce structure, paving the way for further studies on growth regulation leading to precise ultrastructures and structural colors.

---

## [Referee Report · Reviewer #1 (Public Review)]

This article is interested in how butterfly, or more precisely, butterfly wing scale precursor cells, each make precisely patterned ultrastructures made of chitin.

To do this, the authors sought to use the butterfly Parides eurimedes, a papilionid swallowtail, that carries interesting, unusual structures made of 1) vertical ridges, that lack a typical layered stacking arrangement; and 2) deep honeycomb-like pores (rather than. These two features make the organism chosen a good point of comparison with previous studies, including classic papers that relied on electronic microscopy (SEM/TEM), and more recent confocal microscopy studies.

The article shows good microscopy data, including detailed, dense developmental series of staining in the Parides eurimedes model. The mix of cell membrane staining, chitin precursor, and F-actin staining is well utilized and appropriately documented with the held of 3D-SIM, a microscopy technique considered to provide super-resolution (here needed to visualize sub-cellular processes).

The key message from this article is that F-actin filaments are later repurposed, in papilionid butterflies, to finish the patterning of the inter-ridge space, elaborating new structures (this was not observed so far in other studies and organisms). The model proposed in Figure 6 summarized these findings well, with F-actin reshaping itself into a tulip that likely pulls down a chitin disk to form honeycombs. These interpretations of the microscopy data are interesting and novel.

There are two other points of interest, that deserve future investigation:

1. The authors performed immunolocalizations of Arp2 and pharmacological inhibitions of Arp2/3, and found some possible effect on honeycomb lattice development. The inter-ridge region of the butterfly Papilio polytes, which lacks these structures, did not seem to be affected by drug treatments. Effects were time-dependent, which makes sense. These data provide circumstantial evidence that Arp2/3 is involved in the late role of F-actin formation or re-organisation.

2. The authors perform a comparative study in additional papilionids (Fig. 6 in particular). I find these data to be quite limited without a dense sampling, but they are nonetheless interesting and support a second-phase role of F-actin re-organisation.

The article is dense, well produced and succinctly written. I believe this is an interesting and insightful study on a complex process of cell biology, that inspires us to look at basic phenomena in a broader set of organisms.

---

## [Referee Report · Reviewer #2 (Public Review)]

The manuscript by Seah and Saranathan investigates the cell-based growth mechanism of so called honeycomb-structures in the upper lamina of papilionid wing scales by investigating a number of different species. The authors chose Parides eurimedes as a focus species with the developmental pathway of five other papilionid as a comparative backup. Through state-of-the-art microscopy images of different developmental steps, the authors find that the intricate f-actin filaments reorganise, support cuticular discs that template the air holes that form the honeycomb lattice.

The revised manuscript is well written and easy to follow, yet based on a somewhat limited sample size for their focus species, limiting attempts to suppress expression and alter structure shape. I have no further comments.

---

## [Author Response]

The following is the authors’ response to the original reviews.

**Reviewer #1 (Public Review):**
This article is interested in how butterfly, or more precisely, butterfly wing scale precursor cells, each make precisely patterned ultrastructures made of chitin.To do this, the authors sought to use the butterfly Parides eurimedes, a papilionid swallowtail, that carries interesting, unusual structures made of 1) vertical ridges, that lack a typical layered stacking arrangement; and 2) deep honeycomb-like pores. These two features make the organism chosen a good point of comparison with previous studies, including classic papers that relied on electronic microscopy (SEM/TEM), and more recent confocal microscopy studies.The article shows good microscopy data, including detailed, dense developmental series of staining in the Parides eurimedes model. The mix of cell membrane staining, chitin precursor, and F-actin staining is well utilized and appropriately documented with the help of 3D-SIM, a microscopy technique considered to provide super-resolution (here needed to visualize sub-cellular processes).The key message from this article is that F-actin filaments are later repurposed, in papilionid butterflies, to finish the patterning of the inter-ridge space, elaborating new structures (this was not observed so far in other studies and organisms). The model proposed in Figure 6 summarized these findings well, with F-actin reshaping it itself into a tulip that likely pulls down a chitin disk to form honeycombs. These interpretations of the microscopy data are interesting and novel.There are two other points of interest, that deserve future investigation:1. The authors performed immunolocalizations of Arp2 and pharmacological inhibitions of Arp2/3, and found some possible effect on honeycomb lattice development. The inter-ridge region of the butterfly Papilio polytes, which lacks these structures, did not seem to be affected by drug treatments. Effects where time- dependent, which makes sense. These data provide circumstantial evidence that Arp2/3 is involved in the late role of F-actin formation or re-organisation.1. The authors perform a comparative study in additional papilionids (Fig. 6 in particular). I find these data to be quite limited without a dense sampling, but they are nonetheless interesting and support a second-phase role of F-actin re- organisation.The article is dense, well produced and succinctly written. I believe this is an interesting and insightful study on a complex process of cell biology, that inspires us to look at basic phenomena in a broader set of organisms.

We thank the reviewer for the positive appraisal.

**Reviewer #2 (Public Review):**
The manuscript by Seah and Saranathan investigates the cell-based growth mechanism of so called honeycomb-structures in the upper lamina of papilionid wing scales by investigating a number of different species. The authors chose Parides eurimedes as a focus species with the developmental pathway of five other papilionid as a comparative backup. Through state-of-the-art microscopy images of different developmental steps, the author find that the intricate f-actin filaments reorganise, support cuticular discs that template the air holes that form the honeycomb lattice. The manuscript is well written and easy to follow, yet based on a somewhat limited sample size for their focus species, limiting attempts to suppress expression and alter structure shape.The fact that the authors find a novel reorganisation mechanism is exciting and warrants further research, e.g. into the formation of other microscale features or smaller scale structures (e.g. the mentioned gyroid networks).

We thank the reviewer for the positive appraisal.

The authors place their results in the discussion in the light of current literature (although the references could be expanded further to include the breadth of the field). However, the mechanistic explanation completely ignores the mechanical properties of the membranes as an origin of some of the observed phenomena (see McDougal's work for example) and places the occurence of some features into Turing patterns and Ostwald ripening, which I find somewhat unlikely and I suggest that the authors discover this aspects further in the discussion.

We thank the reviewer for these suggestions. We have added more references from the current literature to more accurately reflecting the breadth of the field. McDougal et al. 2021. discuss the nature of biomechanical forces (differential growth and buckling) on the membrane and deposited cuticle shaping the formation of longitudinal ridges. However, here it is the invagination of the plasma membrane bearing the deposited cuticle that is our main concern. Nevertheless, we agree future studies should indeed consider the mechanical properties of the membranes, in addition, to explain some of the observed features. We have clarified this in our discussion.

I have little concerns regarding the experimental approach beyond the somewhat limited sample size. One thing the authors should more clearly mention are the pupation periods for all investigated species as only the periods for two species are named.

Yes, unfortunately, we were only able to obtain pupae with pupation dates for two species. We have clarified this point in the methods.

**Reviewer #1 (Recommendations For The Authors):**
Suggestion for improvement.I recommend adopting a magenta/green (or orange/azure) color scheme to make the figures accessible to most color vision types. This does not require re-doing the figure and could be processed on the rendered JPG/TIF figures with the following procedure :1. open the rendered figures in Photoshop in RGB mode1. go to Channel Mixer1. Select Output Channel : Blue1. set Blue 100%0% and Red 0100%This will change Red to Magenta without affecting luminosity.Similar solutions should be available in other software including GIMP.Of note this is a late fix and ideally, color encoding could be done upstream in the microscopy file extraction software (e.g. Fiji), but I do not think this heavier solution is needed here.

We thank the reviewer for this suggestion. In order to be more inclusive, we have redone the figures and videos in a yellow+magenta color scheme.

**Reviewer #2 (Recommendations For The Authors):**
References: Some literature is missing that could be considered by the authors e.g.
https://doi.org/10.1098/rstb.2020.0505

https://doi.org/10.1101/2023.06.01.542791

https://doi.org/10.1098/rsfs.2011.0082

https://doi.org/10.1557/mrs.2019.21

https://iopscience.iop.org/article/10.1088/2040-8986/aaff39/meta

https://doi.org/10.1364/OE.20.008877

We have added more references as suggested.

Placing the captions next to the figures, particularly in the SI will help accessibility.

We agree. We believe this would be done during article production.

113: chiefly?

We have replaced ‘chiefly’ with ‘focusing mainly on’.

160: how do you know the scales are more scletorized already? Just because it's later in development?

Yes, that is what we are alluding to here. We have made edits to clarify this sentence.

186: Specify sample size.

We have specified the sample size ‘(N = 15)’ here.

309: Multilayered cover scales would be more accurate.

Thanks for the suggestion. We have changed ‘structurally-colored cover scales’ to ‘multilayered cover scales’ as suggested.

Please check the literature list again for accurate references.

Thanks for the suggestion. We have gone through the references and fixed any missing information.